# pyProGA—A PyMOL plugin for protein residue network analysis

**Vladimir Sladek**[1]*, **Yuta Yamamoto**[2], **Ryuhei Harada**[3], **Mitsuo Shoji**[3], **Yasuteru Shigeta**[3], **Vladimir Sladek**[4]

**1** Institute of Chemistry, Slovak Academy of Sciences, Bratislava, Slovakia, **2** Department of Chemistry, Rikkyo University, Nishi-Ikebukuro, Tokyo, Japan, **3** Center for Computational Sciences, University of Tsukuba, Tsukuba, Ibaraki, Japan, **4** Institute of Construction and Architecture, Slovak Academy of Sciences, Bratislava, Slovakia

* sladek.vladimir@savba.sk

## Abstract

The field of protein residue network (PRN) research has brought several useful methods and techniques for structural analysis of proteins and protein complexes. Many of these are ripe and ready to be used by the proteomics community outside of the PRN specialists. In this paper we present software which collects an ensemble of (network) methods tailored towards the analysis of protein-protein interactions (PPI) and/or interactions of proteins with ligands of other type, e.g. nucleic acids, oligosaccharides etc. In parallel, we propose the use of the network differential analysis as a method to identify residues mediating key interactions between proteins. We use a model system, to show that in combination with other, already published methods, also included in pyProGA, it can be used to make such predictions. Such extended repertoire of methods allows to cross-check predictions with other methods as well, as we show here. In addition, the possibility to construct PRN models from various kinds of input is so far a unique asset of our code. One can use structural data as defined in PDB files and/or from data on residue pair interaction energies, either from force-field parameters or fragment molecular orbital (FMO) calculations. pyProGA is a free open-source software available from https://gitlab.com/Vlado_S/pyproga.

**Data Availability Statement:** pyProGA is a free open-source software available from https://gitlab.com/Vlado_S/pyproga.

**Funding:** This research was supported by the Slovak Research and Development Agency (APVV,

## Introduction

The science of protein residue network (PRN) models is being developed over almost three decades. This may indicate that for the community it is appealing to have exact, qualitative and quantitative protein models to study their structure, topology and dynamics. Let us first, without claim of completeness, establish a framework for the current development stage of PRN models in order to understand why we felt the necessity to build pyProGA.

As the name suggest, PRN models are network models, in which the constituents are protein residues (represented as vertices/nodes) and the interactions between them (represented as edges) [1]. There is a different genre of PIN—Protein Interaction Network research in which the protein is the node and interactions between proteins are studied at molecular, but not at residue level [2–4]. These are not the subject of this work.

https://www.apvv.sk/?lang=en) in the form of a grant awarded to VS (APVV-19-0376), the Scientific Grant Agency of the Ministry of Education, Science, Research and Sport of the Slovak Republic and The Slovak Academy of Sciences (VEGA, https://www.minedu.sk/vedecka-grantova-agentura-msvvas-sr-a-sav-vega/) in the form of grants awarded to VS (VEGA-2/0031/19, VEGA-2/0061/20), the MEXT Quantum Leap Flagship Program (MEXT Q-LEAP, https://www.jst.go.jp/stpp/q-leap/en/index.html) in the form of a grant awarded to YS (JPMXS0120330644), and in part by the Japan Agency for Medical Research and Development (AMED, https://www.amed.go.jp/en/) in the form of a grant awarded to YS (JP20ae0101047h0001). Part of the calculations were done at the Computing Center of SAS using the infrastructure from Project Nos. ITMS 26230120002 and 26210120002, supported by the Research and Development Operational Program funded by the ERDF. The funders had no role in study design, data collection and analysis, decision to publish, or preparation of the manuscript.

**Competing interests:** The authors have declared that no competing interests exist.

The mathematical formalism of network science comes from graph theory [5]. One of the earlier and influential application to protein residues was by Kannan and Vishveshwara [6], who attempted to identify side-chain clusters by graph spectral methods. Others have followed to build PRN models to study the network-like structure of proteins [7]. The fact that these protein models obey hierarchical structure, as often found in other, unrelated networks, e.g. small-world character [8], was soon discovered [9–12]. PRNs were used for identification of efficient communication pathways and used to explain allostery effects in proteins and protein clusters [13–15] and were shown to be related to some physical phenomena [16, 17]. An interesting topic is the study of Energy Exchange Networks for proteins as done by Leitner [18] Yamato and others [19–21]. Such and similar models evaluate heat/energy transfer, typically quantities used in continuum theories, over a discretized irregular mesh represented by the network.

The focus on dynamical network models, or rather the dynamics of PRN was explored as well [22]. VanWarth *et al.* [23] examined dynamic models of allostery, see also [24]. Conformational changes in PRNs were elucidated in several works [25–27] The field of protein folding also adopted ideas from PRN models [28, 29]. There are software options for such dynamic PRN models such as gRINN [30] or RIP-MD [31]. We do not include PRN dynamics in pyProGA, as PyMOL is not particularly suited for analysing molecular dynamics trajectories, and, as mentioned before, the static models have applications in their own right.

It was discovered that these models are more than useful tools to find interesting clusters of amino acids or communication pathways [32]. Estrada [33] discovered that there was some generality in hierarchy of the network and organisation of residues, see also [34]. As the PRNs were getting more widespread recognition, a natural desire to standardise their creation emerged. Viloria *et al.* [35] attempted to define an optimal cut-off distance between residues to form sensible contact based PRNs. It should be noted that the majority of the models were based on some distance criteria (either closest atoms, centre of mass or C$\alpha$ distances). The work of Vijayabaskar *et al.* [36] introduced PRNs based on pair interaction energies (PIE) between the residues. The PIEs in that work were from parametrised force field pair potentials (as used in molecular dynamics). One of our works [37] attempted to explore differences between PRNs based on distance (D-PRN) criteria and those based on interaction energies (PIE-PRN). We found that in D-PRNs the node hierarchy, i.e. ranking by some centrality measure, tends to be more sensitive to cut-off criteria. Similar behaviour can be observed in case of community structure. Recently, Yao *et al.* [38] sought to establish a more concise framework for using D- and PIE-PRNs. They have found that certain structure equivalence of these two models is observed, when the distance cut-off between the closest non-hydrogen atoms in two residues is some 4.5Å and the energy cut-off is about $kT$, $k$ being the Boltzmann constant, $T$ the temperature.

Practical applications for PRNs were also found. Haratipour *et al.* [39] characterised some typical structural unit *via* quantitative analysis of PRNs such as centrality calculation etc. Capriotti *et al.* [40] examined protein stability upon single point mutations. We shed some light on community structure in PRNs and the relation to protein-peptide binding analysis [37]. Estrada [41] recently published a topological analysis of the SARS CoV-2 protease M$^{pro}$ using a residue network model. Not long ago the binding of nucleic acids to proteins was examined by Miao and Westhof [42]. It should be noted that while we saw these practical applications, most of them were carried out by researchers in the field of protein residue networks, i.e. mostly using original "in-house" code. This presents an obstacle, to some degree, for a "trickle down" effect by which also non-experts in PRN research could use these methods and algorithms in their labs, albeit there is some current progress at this front as well. Aydinkal *et al.* [43] developed the ProSNEx web server. The NAPS web tool by Chakrabarty and Parekh

[44] is currently available. Felline *et al.* [45] offered their webPSN v2.0 web server including their elastic network model for normal mode analysis [46]. RING 2.0 [47] is another web-server capable of identifying different kinds of covalent and non-covalent interactions in PDB structures. The tool by Yan *et al.* [48] is dedicated to find different kinds of functional residues. xPyder by Pasi *et al.* [49] is a plugin for PyMOL 1.5. We recommend also the work by Shcherbinin and Veselovsky [50] to look for an extensive overview in this field.

The reasons for us to develop pyProGA—python Protein Graph Analyser are multifarious. Most importantly, the fact that pyProGA is a PyMOL [51] plugin is in our view a significant plus. The PyMOL package is widely used in molecular research [52] and thus enjoys a large degree of familiarity in the chemical community. Additionally, PyMOL offers a plethora of different visualization options enabling us to utilise them for display of the PRNs themselves or the results of various analyses. We also view the open source nature of PyMOL and pyProGA as advantageous, as it facilitates modular extension of its' contents. Hence, we hope to have created a platform that may be adopted in other research groups to build upon. We have included basic network analytic algorithms for node and edge centralities, network partition, etc. We include the possibility to calculate the residue folding degree [53, 54]. Strong accent was given to a user-friendly interface with accent to powerful protein-protein interaction analytical techniques, sensible visualization of results, visualization of the networks themselves and good documentation in the manual. A detailed introduction of graph theory with its terminology and use in network science is beyond the scope of this paper. Nonetheless, we attempts to provide such introduction with explanations in the accompanying manual. PyProGA was designed to be used on both D-PRN and PIE-PRN protein models, where the PIE-PRNs can be based on either Fragment Molecular Orbital (FMO) results [55] or force-field PIEs. The same network analysis can be performed on all types of PRNs, facilitating direct comparison of results, which is unique.

## Methods

The Methods section will describe the structure and functionality of pyProGA. Fig 1 is a summarisation of this. In the Discussion part we will provide some use examples. We are not going to deal with technicalities such as the installation process or step-by-step examples, which are detailed in the manual. Nonetheless, this is the basic software framework used for

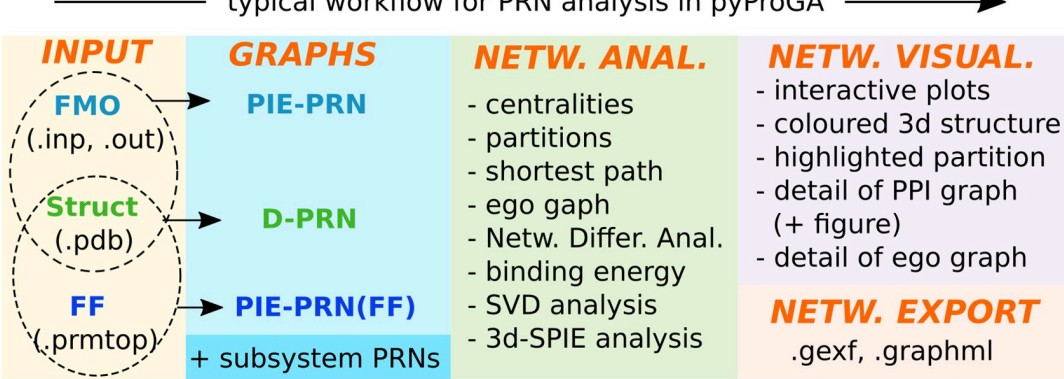

**Fig 1. Basic overview of pyProGA functionality.** Input requirements are on the left, data structure in the centre and various analytical and output options are shown. The subsystem analysis requires the same kind of files to be loaded for the monomers as was for the super-system (dimer).

developing pyProGA; it was written in Python 3.7 using the PyQt 5 GUI toolkit which works well and is recommended with PyMOL 2.4 (current distribution version).

## Input and data structures

In order to use this plugin a minimum requirement is to load a .pdb file of the molecular system that will be studied. We use the Bio.PDB parser [56] to accomplish this. From this we can directly proceed to create a distance based residue network, D-PRN. In this case each residue as defined in the .pdb file is one node in the network. In order to create a pair interaction energy based network (PIE-PRN), we must additionally supply either an output from an Gamess FMO (Fragment Molecular Orbital) calculation or an Amber .prmtop file.

Gamess [57, 58] is a quantum chemistry package including the FMO method [59, 60]. There are various orders of approximation in FMO [61], however, for our purpose the two-body approximation is sufficient, as we are mostly interested in the pair interactions, sometimes called Inter Fragment Interaction Energies (IFIEs). For objectivity sake we note that there are other programs facilitating FMO calculations, most notably Abinit MP [62] and PAICS [63]. pyProGA is currently not able to parse their outputs. Nevertheless, we are open to include such support if demand arises, albeit direct cooperation with the developers will probably be necessary to assure bug-free functionality. In case of PIE-PRNs from FMO calculations the nodes correspond to FMO fragments. These, in general, are not equal to amino acid residues as per PDB standards (the C,O atoms assignment is shifted by one to the neighbouring residue). This is because fractioning at the peptide bond led to accuracy issues in FMO [64]. For covalently non-bonded and non amino acid residues the correspondence between fragments and residues can exist. Very recent research [65] promises correction of this minor drawback when density fitted tight binding (DFTB) is used in FMO. Additionally, we recommend the use of some solvation model with FMO [66, 67]. The use of PIEDA (PIE Decomposition Analysis) is supported in pyProGA [68–70]. We recommend the Facio toolkit (available from http://zzzfelis.sakura.ne.jp/index.html) [71] to work with FMO fragments.

For PIE-PRN models based on PIE values coming from force field (FF) data one must supply a Amber .prmtop file [72]. For simplicity, let's designate these as PIE-PRN(FF). As of now we do not support other formats of parsing FF parameters to pyProGA. At this point we should emphasize that we need the FF parameters only to calculate the PIEs in the structure as is in the .pdb file. We do not need any MD trajectory nor any other files related to other specific MD software.

The main data structure/object in pyProGA holding the network topology plus some additional data is a graph $G$. A graph is a collection of a set of vertices and edges, $G = (V, \mathcal{E})$. The graph object is as standardised by the NetworkX (v. 2.5) python module [73]. A undirected graph is most suited for our PRN models. The graph $G$ is created when the user sets the edge acceptance criteria. In case of D-PRNs it is the cut-off value $R_\mathrm{lim}$ for the centre of mass distance of two residues/nodes, $i, j$, in the structure. For any PIE-PRN the primary and only mandatory criterion is the PIE cut-off value $E_\mathrm{lim}$. $R_\mathrm{lim}$ is a secondary optional criterion. Finer specification of PIE energy terms is possible; $E_\mathrm{tot}$ i.e. the total pair interaction energy is available for any PIE-PRN. If PIEDA was used, then any of its terms can be chosen. One has to bear in mind that not all PIEDA terms necessarily acquire negative values, which is a consequence of the physical definition of these terms. For PIE-PRN(FF) one can use, in addition to $E_\mathrm{tot}$, also the electrostatic or the van der Waals component. The user sets how the edge weight relates to the energy (for PIE-PRNs) or distance (for D-PRNs). In general, the default options that $w_{ij} \simeq |E_{ij}|^{-1}$ or $w_{ij} \simeq R_{ij}$ are advised, see [37]. The edge importance is always the inverse of the edge weight.

## Simple network analysis

pyProGA enables the user to analyse the PRNs in terms of node and edge centralities. We rely on the NetworkX modules, except for the weighted efficiency centrality [74] for which we use our own code [75], together with the evaluation of the weighted global and local efficiencies [76]. For now, we include several well known centralities; closeness centrality, betweenness centrality, degree centrality and the current flow betweenness centrality. In addition, we define the total energy centrality $C_k^{E_{tot}}$. It is a variation of the degree centrality, where we sum total energy terms $E_{tot}$ for all edges $\mathcal{E}_{i,k}$ adjacent/connected to the node $k$ in $G$. A node with low (more negative) $C_k^{E_{tot}}$ is considered to be "strongly bound" in $G$—an interpretation often used for FMO based studies [77, 78].

The results can be viewed in several forms. PyProGA creates a new PyMOL object after calculating any of the centralities (one object for each centrality type and network type). A colour gradient is applied to the atoms corresponding to the nodes (residues) to indicate the magnitude of the centralities. This facilitates quick visual inspection of results. In addition, a bar plot of the centralities per node can be examined in an additional window. The bar plot is clickable, hence centrality values for a particular node are easy to acquire. A histogram is also always depicted. All the plots can be further manipulated by standard Matplotlib functions. Finally, the results are saved in text format as well.

In addition to centralities, graph partitions can be evaluated as well. As of now, we support the calculation of spectral clustering [6, 79] and modularity based communities *via* the Louvain algorithm [80]. Also here a new PyMOL object is created to show the colour-defined partitions. Also, each of the partitions can be highlighted individually. Text files with results are saved.

Finally, pyProGA enables the search for the shortest path connecting any pair of nodes. This function can be useful in specific cases.

The edge properties, such as energy $E_{ij}$ and its components if PIEDA was used, node distance $R_{ij}$, and several other, can be viewed in an interactive 2D map.

## Network analysis of supramolecular systems

In this section we are going to discuss the analysis of supramolecular systems, which the user deliberately sets up for this purpose. By a supramolecular system we understand any system of at least two molecules. For our purpose we are going to limit ourselves to systems, which may be thought of as dimers, i.e. containing subsystems A and B. In such case we create additional graphs/networks $G_{A(B)}$ representing the monomers. They may be e.g. both proteins forming a protein complex or one a protein and second a ligand. To create $G_{A(B)}$ one must load the same kind of input as for the dimer graph $G$, see Fig 1.

Recalling that a graph is a collection of a set of vertices and edges, $G = (V, \mathcal{E})$, then, in general for two graphs $G'$ and $G''$, their union is defined as $G' \cup G'' : (V' \cup V'', \mathcal{E}' \cup \mathcal{E}'')$. Hence, we can formally write that $G = G_A \cup G_B \cup G_{PPI}$. Therefore when the user defines the graphs $G_{A(B)}$, both sub-graphs of $G$, we can define the bipartite graph $G_{PPI}$. This graph describes the topology of the interactions in the protein-protein interface (PPI). Having this formalism allows for further investigation of the PPI interactions. The straight-forward step is to save a figure of $G_{PPI}$ in pyProGA.

**FMO binding energy.**  Fedorov and Kitaura [81] have proposed a way how to divide the total (supramolecular) interaction energy into contributions of each fragment as used in the FMO calculation. Such energy is then called the binding energy and it is not the same as a simple sum of the PIEs for each node like $C_k^{E_{tot}}$. We can think of it as a measure of how much each

fragment in one monomer, A or B, contributes to the binding of the second monomer. The assignment of the binding energies can be done either symmetrically to fragments in both A and B or asymmetrically to either A or B. In this way one can examine dominant fragments contributing to the stabilization of the dimer. This method itself is not based on a typical network analysis, yet we find it very useful, hence it is included in pyProGA. The results are, as usually, saved in a text file and displayed in the PyMOL window where the residues are coloured by a colour map scale reflecting the magnitude of the binding energy. Clickable Matplotlib bar plots and histograms are drawn on demand.

**3D SPIE.** Lim *et al.* [82] have proposed the 3D Scattered PIE as a useful way to investigate PPI interfaces. In essence it is a 3D scatter plot where the $G_{PPI}$ nodes from monomer A and B are on the two independent axes, and the data points in the third direction represent the magnitude of the interaction energy. We can add a colour palette to differentiate the repulsive and attractive interactions. pyProGA enables the creation of such plots for any of the energy components used for the construction of the graph.

**SVD analysis of PPI.** Tanaka *et al.* [78, 83] have shown how the singular value decomposition (SVD) technique can be used to investigate protein-protein interfaces. The authors describe their analysis as "network-like". In fact, we can argue that it is a network analysis, considering that they use a specific part of the weighted adjacency matrix. Specifically, they use the block(s) corresponding to the PPI interactions, see the yellow rectangles in Fig 2. From the SVD procedure they identify dominant motifs in the PP interface, similarly to a principal component (PC) analysis. This way, one can asses the importance of individual nodes/residues and/or clusters of nodes. This technique is implemented in pyProGA. Moreover, we provide an adapted version of it, where we subject the full adjacency matrix of the $G_{PPI}$ to the SVD algorithm. There is no inherent advantage to this, except that in specific cases one can deal with smaller matrices and hence the interpretation can be somewhat more straightforward. To see the difference, let us consider this example; the original method by Tanaka *et al.* [78] deals with a matrix of the size $N_A \times N_B$ if we assume $N_A$ and $N_B$ nodes in monomers A and B, respectively. If the proteins are large, then this matrix will be large as well and contain nodes that do not contribute to the PPI in any way. In such case it may prove efficient to use the square matrix $\mathbf{A}_{PPI}$, as this is a representation of $G_{PPI}$, which contains only $N_{PPI}$ nodes constituting the PPI, see Fig 2. In general, the matrix $\mathbf{M}$ subjected to SVD will be of shape $n \times m$, where $n$ and $m$ may or may not equal, depending on which matrix is analysed.

$$\mathbf{M} = \mathbf{U}\mathbf{\Sigma}\mathbf{V}^* \tag{1}$$

$$\tilde{\mathbf{M}}^{(r)} = \sum_{i=1}^{r} \sigma_i \mathbf{u}_i \otimes \mathbf{v}_i^* \tag{2}$$

The SVD method allows to rewrite the matrix $\mathbf{M}$ in terms of three special matrices as in Eq 1. Therein $\mathbf{U} \in \mathbb{R}^{n \times k}$, $\mathbf{\Sigma} \in \mathbb{R}^{k \times k}$ and $\mathbf{V}^* \in \mathbb{R}^{k \times m}$, where $k = \min(n, m)$. The columns of $\mathbf{U}$, $\{\mathbf{u}_{i=1...k}\}$, are called left singular vectors of $\mathbf{M}$ and the rows of $\mathbf{V}^*$, $\{\mathbf{v}_{i=1...k}^*\}$, (also columns of $\mathbf{V}$) are the right singular vectors. The vectors $\{\mathbf{u}_i\}$ are orthogonal and so are the vectors $\{\mathbf{v}_i^*\}$. The matrix $\mathbf{\Sigma}$ is a diagonal matrix, and its diagonal elements $\sigma_{i=1...k}$ are called the singular values. Typically, they are ordered in such way that $\sigma_i \geq \sigma_{i+1}$. This allows to approximate $\mathbf{M}$ to a $r$-th degree as in Eq 2 *via* the sum of the outer (dyadic) products of the $r$ leading vectors of $\mathbf{U}$ and $\mathbf{V}^*$. Since the lengths of $\mathbf{u}_i$ and $\mathbf{v}_i^*$ are $n$ and $m$, respectively, the matrix $\tilde{\mathbf{M}}^{(r)}$ will be of shape $n \times m$, as is $\mathbf{M}$. For $r = k$ we get Eq 1, *ergo* $\tilde{\mathbf{M}}^{(r=k)} = \mathbf{M}$. A way to judge the relative degree to

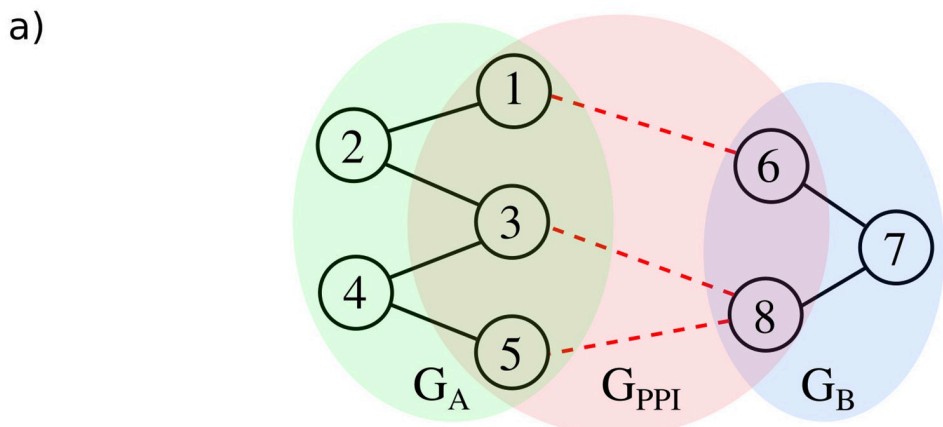

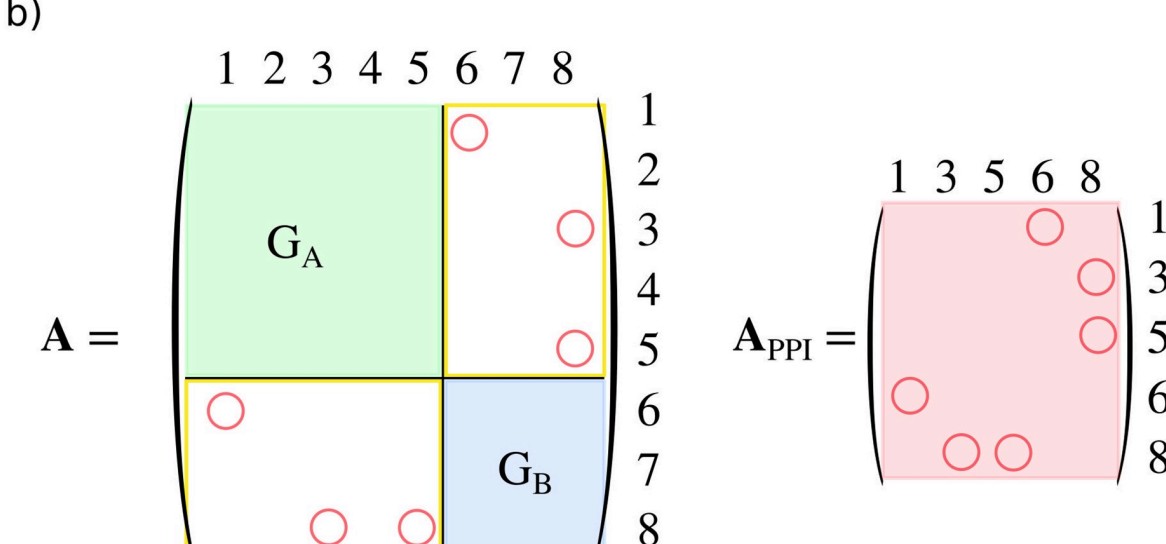

**Fig 2. Subunits of the network.** The PRN of the supersystem consisting of monomers *A* and *B* and the protein-protein interface (PPI): $G = G_A \cup G_B \cup G_{PPI}$. $G_A$ and $G_B$ contain all nodes and edges within protein *A* and *B*, respectively. a) $G_{PPI}$ contains vertices from both monomers, but only those vertices, which have at least one edge connecting it to a vertex from the other monomer. The edges in $G_{PPI}$ are only those which connect the subgraphs $G_A$ and $G_B$ (dashed lines) and no edges within the subgraphs $G_A$, $G_B$. Hence, $G_{PPI}$ is a bipartite graph. Part b) shows how blocks in the adjacency matrix **A** correspond to parts of *G* and the adjacency matrix $\mathbf{A}_{PPI}$ of $G_{PPI}$ (circles indicating edges in $G_{PPI}$).

which $\tilde{\mathbf{M}}^{(r)}$ approximates **M** is by the factor $f^{(r)}$ defined as

$$f^{(r)} = \sum_{i=1}^{r} \sigma_i / \sum_{i=1}^{k} \sigma_i \; ; f^{(r=k)} = 1 \tag{3}$$

The way how this method is included in pyProGA allows for a simple choice of whether we intent to look at attractive, repulsive or both types of PPI interactions. The original authors seem not to entertain these possibilities and we assume that they analysed both types at once. Additionally, we may choose whether we look at motifs in the monomer A or B. Within each *i*-th motif (see Eq 2) we look at the dominant nodes, and we automatically see their strongest interacting partners (connected nodes) from the other monomer.

**Network Differential Analysis (NDA).**   We have briefly outlined how one can use common centrality algorithms to rank the nodes in a PRN. If ve have also graphs for the monomers, we can examine how the edges connecting them affect the centrality ranking of each node. We do this by calculating the node centralities $C_k$ in $G$, the PRN of the complex, as well as in graphs $G_A$ and $G_B$. The difference of the centralities can indicate which nodes gain higher rank when edges forming the PPI are present. In pyProGA we call it Network Differential Analysis (NDA). However, this procedure would lead us to compare centralities between graphs of different size (number of nodes). Consequently, since centrality measures are often not directly transferable/comparable between different graphs, the ranking may not be always reliable. To overcome this issue we define the centrality difference as

$$\Delta C_k = C_k(G) - C_k(G_{A \cup B}) \tag{4}$$

This is the default way to calculate NDA centrality in pyProGA. The graph $G_{A \cup B}$ has all nodes of $G$ and only the edges present in the graph $G_{\mathrm{PPI}}$ are missing.

## Results

We use the *de novo* synthesized TIM barrel protein [84] (PDB ID: 5BVL) to demonstrate selected features of pyProGA. The structure in the pdb file is four-fold symmetrical and has missing residue number 17. This is just behind the first helix. We took advantage of this as it enables us to treat this system formally as a complex of one helix (monomer A) with the remainder of the protein (monomer B). All data to reproduce this test are provided with the distribution of pyProGA. Please do read the S1 File for much more in-depth comments.

We load the whole system AB *via* the pyProGA interface and apply the edge acceptance criterion $E_{\mathrm{tot}} \leq -1 \mathrm{kcal\,mol}^{-1}$ to all types of bonds (incl. peptide) with the remaining settings kept at default. In the first step we calculate the total energy centrality $C_k^{E_{\mathrm{tot}}}$ and the efficiency centrality $C_k^{\mathrm{eff}}$ in the dimer PRN $G$, see S4a and S5a Figs in S1 File. We see that these are rather inconclusive if we are interested in how the monomers A and B interact. Therefore we load the monomer data into pyProGA and construct the monomer PRNs $G_A$ and $G_B$ (same rules as for $G$ apply). This allows us to calculate the centrality differences $\Delta C_k^{E_{\mathrm{tot}}}$ and $\Delta C_k^{\mathrm{eff}}$ according to Eq 4, see Fig 3 and S4b and S5b Figs in S1 File. S6 and S7 Figs in S1 File show the centralities depicted as bar plots for each node/residuum which make it easy to identify high-scoring nodes. The residues Lys2 and Asp29 can be readily picked out as important for the binding of the two monomers. Similarly, albeit with somewhat lower centrality score, residues Lys169 and Glu15 are seen as important. In general, we can conclude that both $\Delta C_k^{E_{\mathrm{tot}}}$ and $\Delta C_k^{\mathrm{eff}}$ are useful in identifying key nodes with respect to the dimer stabilization. There are some differences, mainly because the efficiency centrality score accounts also for how central given residues are within the monomers, es exemplified on Asp3 from monomer A in S7 Fig in S1 File. The fact that Asp3 is covalently bound (very low edge weight) to Lys2 also adds to its higher $\Delta C_k^{\mathrm{eff}}$ rank.

In the next step we employ the SVD method to investigate the attractive interactions in the bipartite PPI graph $G_{\mathrm{PPI}}$. The particular implementation in pyProGA facilitates us to investigate either $G_{\mathrm{PPI}}$ containing only attractive, only repulsive or both kinds of interactions. In any case, the matrix, which is subjected to the SVD method, contains the corresponding $E_{\mathrm{tot}}$ energies. The results in Fig 4 reflect this. We see that the first/dominant principal coordinates (PC) found by the algorithm (a.k.a. motifs) do correlate strongly to the magnitude of the interaction energy. If we define a measure $f^{(r)}$ of convergence of the matrix approximation, see Eqs 3 and 2, then the first two PCs accounts for about 40% of all terms in the sum. The first four PCs

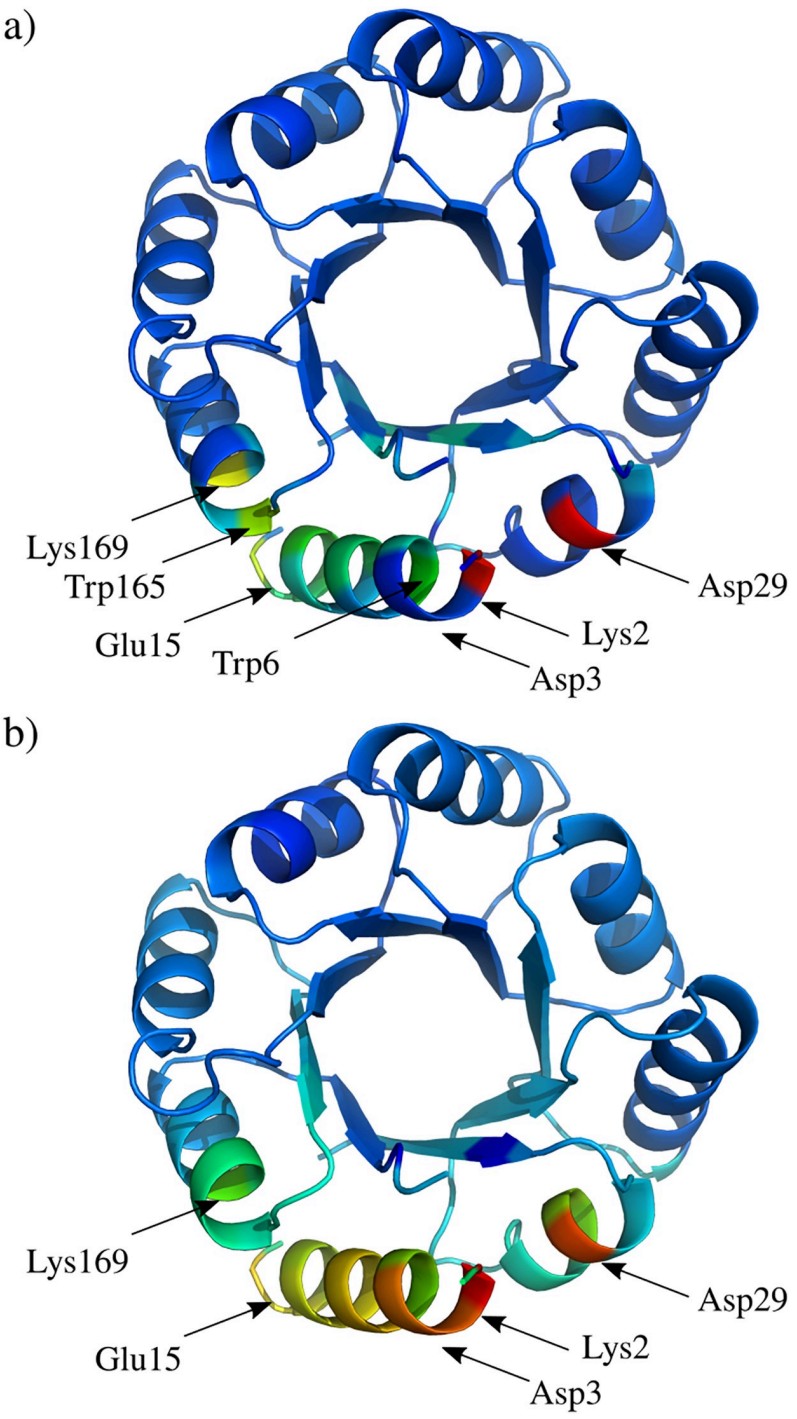

**Fig 3. Network Differential Analysis (NDA).** The protein structure coloured to represent the magnitude of the centrality for each residuum. The PyMOL colour palette `rainbow` is used (red colour for high centrality, blue for low values). In a) $\Delta C_k^{E_{tot}}$ is shown and in b) we show $\Delta C_k^{eff}$ results. Selected high scoring residues are labelled.

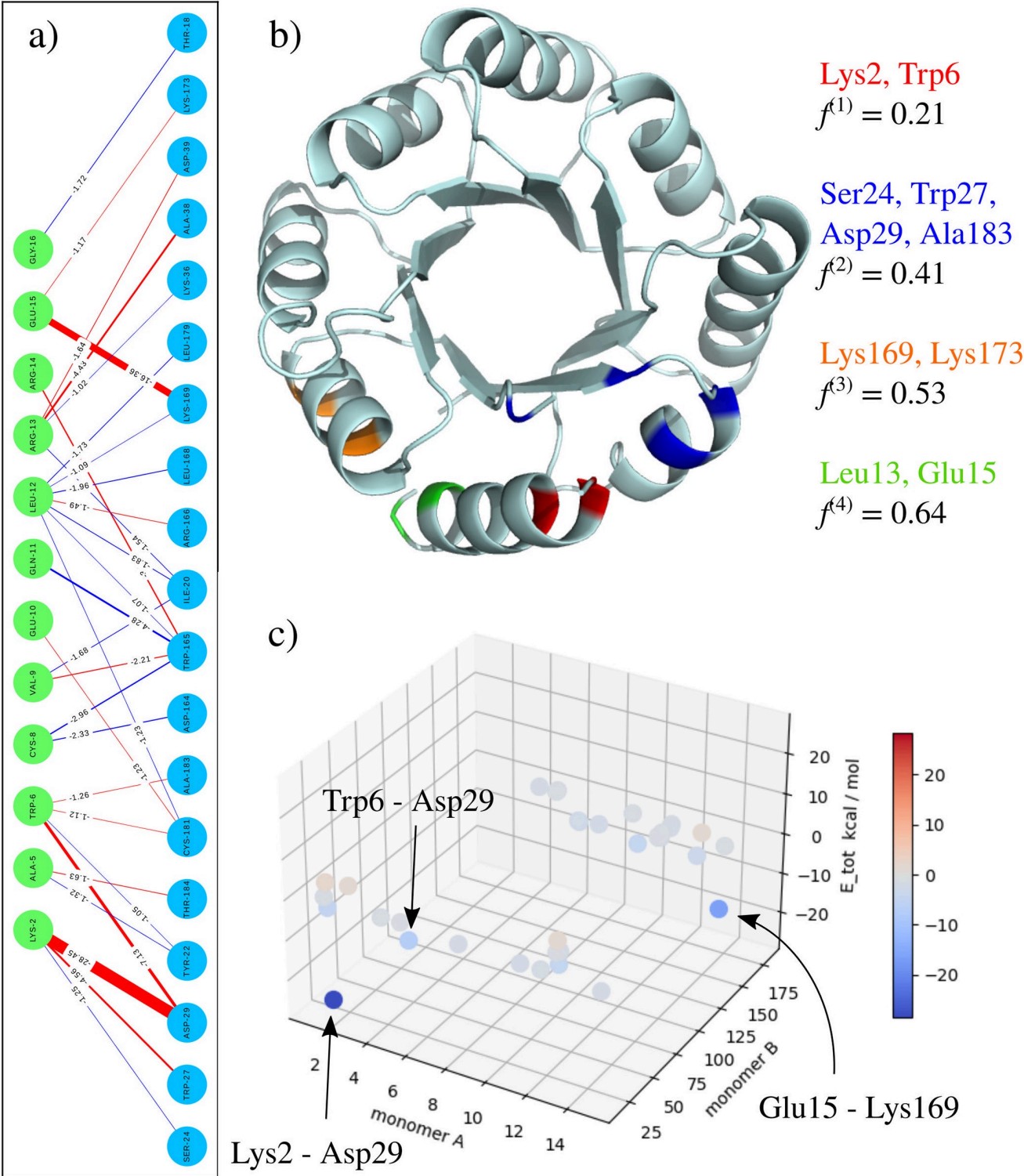

**Fig 4. PPI analysis in pyProGA.** a) The attractive $G_{PPI}$. Green monomer A, blue B. Thickness of edge corresponds to interaction strength, colour to character; red for prevailing electrostatic, blue for dispersion, see [85]. b) First four most dominant principal coordinates (interaction motifs in the PPI) as identified by the SVD analysis of $G_{PPI}$. Colour coded assignment of residues to PC. Factor $f^{(r)}$ is defined by Eq 3. c) 3D-SPIE plot helps to identify strongest attractive and repulsive interactions between monomers A and B. More details in S1 File.

make up to 64% and capture all the dominant PIEs as seen in Fig 4a (for details see also S8–S10 Figs in S1 File).

In general, we can conclude that the results from the SVD method correlate well with the prediction from the $\Delta C_k^{E_{\text{tot}}}$ analysis. Minor discrepancies from the $\Delta C_k^{\text{eff}}$ can be observed. These are explained by the fact that the methods centred purely on $E_{\text{tot}}$ do not account for the network character of the PRN model, see S5 Fig in S1 File for detailed explanation.

Lastly, we look at the community structure of our PIE-PRN model. The concept of finding communities is to identify groups of vertices, which, broadly speaking, are more densely connected than the average edge density in the whole graph. The implicit assumption is that such communities (partitions in general) form components of the graph within which nodes "have something in common", see [80]. Our analysis revealed fourteen communities in $G$, see S11 Fig in S1 File. Quickly it becomes evident that the community structure does, to a large degree, reveal secondary structure elements such as helices etc. This was found also previously, e.g. [37], and is not surprising, as secondary structure is formed due to relatively strong interactions, often with a characteristic motif like in helices. However, the correspondence of a community to a secondary structure is not absolute and in several occasions residues from distinct secondary structure elements belong to one community. This enables us to focus attention to interactions responsible for tertiary structure. pyProGA has several ways in which the network partitions can be visualized, see S11 Fig in S1 File.

Finally, we note that pyProGA can write graphs in .gexf and .graphml format for further freedom in visualising your results in other programs, e.g. Gephi which was used to prepare S12 Fig in S1 File [86].

Most of the methods implemented in pyProGA were published elsewhere (see the Introduction and Methods sections). The purpose of this paper in not to reevaluate their correctness. The network differential analysis (NDA) is new in this respect, hence we validated its predictions against the results of the other methods. We see that they largely agree. Nonetheless, in order to provide comparison to experimental data, we add a short discussion of an system that will be subjected to a separate more in depth study in the future. The system is the protein complex of the human cytomegalovirus protein UL141 with the TRAIL-R2 receptor protein (death receptor DR5) [87]. The authors of that work have performed mutations of the protein at specific sites to asses the effect of such mutation on the stability of the protein complex, from which one can in hindsight asses the importance of the native amino acid in the binding. In some cases they mutated isolated amino acids and in some cases groups of amino acids. We calculated FMO/DFTB PIEs (just like for the TIM barrel protein) and performed the NDA algorithm to obtain the differential centralities $\Delta C_k^{\text{eff}}$. The NDA predictions and the summary of the experimental conclusions are presented in Fig 5. On the first glance, we can tell that several of the residues the mutation of which causes loss of stability of the protein complex can be picked out by the $\Delta C_k^{\text{eff}}$ results. Notably Glu151, Arg133 attain high scores and the mutation of these amino acids seems to prevent the complex formation. However, a simple quantitative relationship cannot be drawn from this data, e.g. the combined mutation of Tyr103/Asn134 disrupts the binding as well, yet these residues score lower than e.g. Asp109, the mutation of which (together with Glu78) causes only a 10-fold lower binding rate. But still, all the residues which are predicted by SPR as important for the complex stability attain relatively high score in our NDA analysis. In contrast, the residues which have lower effect on the binding stability (Pro150, Met152, Lys155) attain also low $\Delta C_k^{\text{eff}}$ values. Combined mutations like the one of Val167/Trp173/Val179 are harder to relate to the $\Delta C_k^{\text{eff}}$ values. Nonetheless on the qualitative level the NDA predictions are very nicely relatable to the experimental data. One must bear in mind that in this preliminary study no dynamic effects were considered.

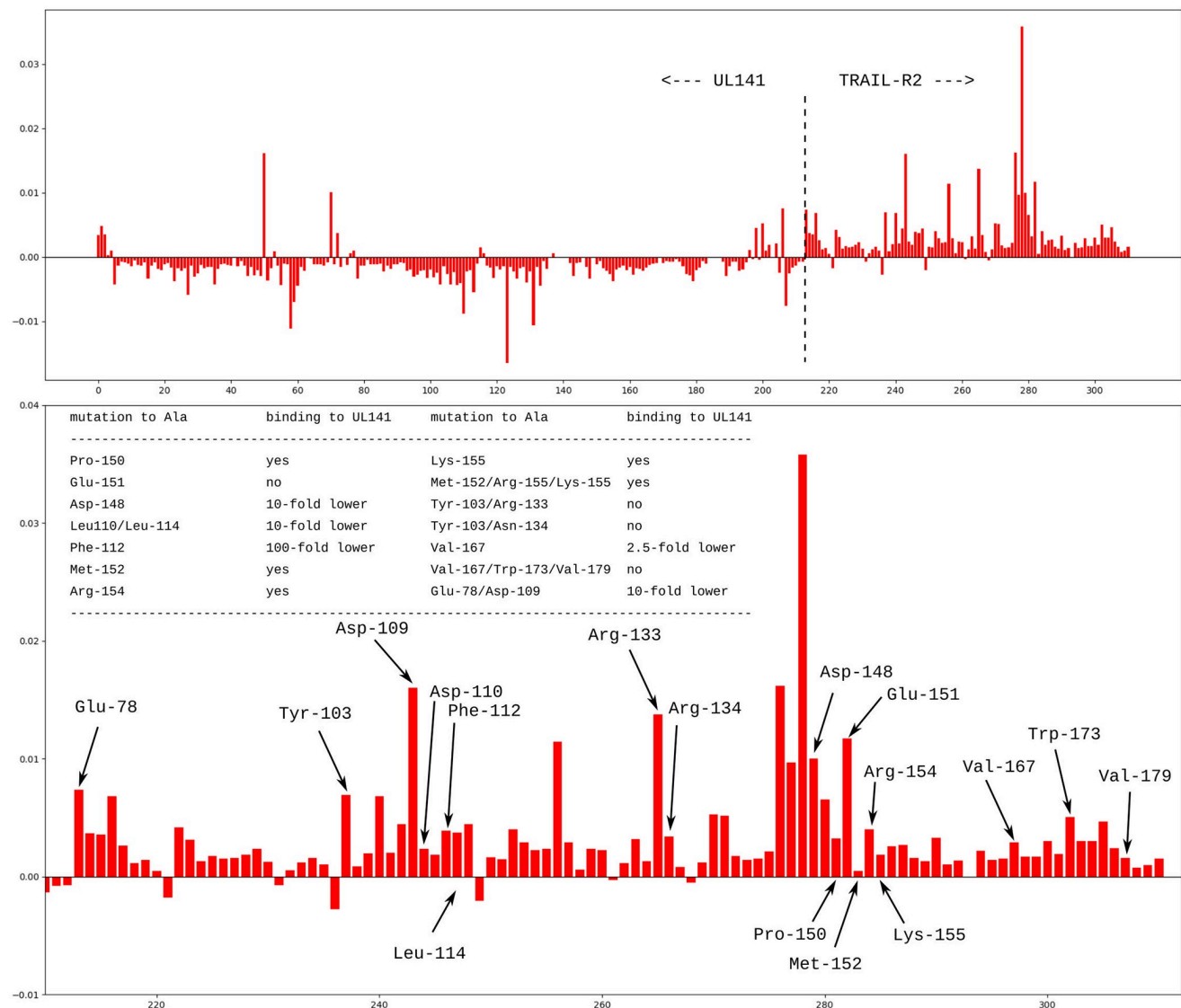

**Fig 5. NDA predictions and experimental data.** The NDA $\Delta C_k^{\text{eff}}$ scores for the residues in the UL141...TRAIL-R2 complex based on an FMO/DFTB calculation. The top panel shows the scores of residues in both proteins (the border is depicted by the dashed line) and the bottom is a detailed plot for the residues of the TRAIL-R2 protein. The table contains experimental SPR (surface plasmon resonance) data published elsewhere [87]. Specific site mutations in the TRAIL-R2 protein to alanine (so called alanine scan), resulted in altered stability of the complex. The NDA bars corresponding to the residues that were mutated in the SPR experiments are labelled.

Also, the SPR protocol cannot determine whether the loss in binding affinity is solely due to the loss of interactions of the mutated amino acid or other effects. Most notably, the structure of the mutated protein may be quite different from the native geometry. Our analysis at this point considers only the native complex structure. We hope to elucidate the effects of dynamics in a future work.

## Conclusion

This paper introduces pyProGA as a tool for analysis of static protein residue networks. The use example exhibits a subset of its capabilities, yet we believe the investigation of protein-protein interfaces is amongst it strongest features. The usefulness of this software is not limited to

protein-protein interactions. Interactions of proteins with other kinds of ligands, such as nucleic acids, oligosaccharides etc. can be studied equally well. The results achieved by various approaches, be it network differential analysis of $\Delta C_k^{E_{tot}}$ and $\Delta C_k^{eff}$, the FMO binding analysis (see S13 Fig in S1 File) and the identified principal components *via* the SVD method are to a large degree self consistent. It seems that the relationship of high and low $\Delta C_k^{eff}$ score to experimental data can be established at a qualitative level even from static models.

There are several ways how to inspect the results visually directly in pyProGA windows and/or in the PyMOL main window. This should provide sufficient flexibility for achieving a close-to-optimal solution for users. In addition, most of the results are also written in text format, which gives even more freedom for alternate modes of representation. As to the depiction of the network(s), we provide 3D models which can be overlaid with any of the objects in PyMOL and the option to save the graph (with node and edge attributes) to standard formats (.gexf, .graphml). We hope to have created a useful tool, which will open the way for PRN models to reach a wider part of the proteomics community.

## Supporting information

**S1 File. The file contains additional figures with extended captions.**
(PDF)

## Author Contributions

**Conceptualization:** Vladimir Sladek, Vladimir Sladek.

**Data curation:** Vladimir Sladek, Yuta Yamamoto, Ryuhei Harada, Mitsuo Shoji.

**Formal analysis:** Yuta Yamamoto, Ryuhei Harada, Mitsuo Shoji, Yasuteru Shigeta, Vladimir Sladek.

**Funding acquisition:** Vladimir Sladek.

**Investigation:** Yasuteru Shigeta.

**Methodology:** Vladimir Sladek.

**Project administration:** Yasuteru Shigeta.

**Software:** Vladimir Sladek.

**Supervision:** Ryuhei Harada.

**Validation:** Vladimir Sladek.

**Visualization:** Vladimir Sladek.

**Writing – original draft:** Vladimir Sladek.

**Writing – review & editing:** Yuta Yamamoto, Ryuhei Harada, Mitsuo Shoji, Yasuteru Shigeta, Vladimir Sladek.

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
