## [Decision Letter · Decision Letter 0]

23 Jun 2021

PONE-D-21-15353

pyProGA - A PyMOL plugin for protein residue network analysis

PLOS ONE

Dear Dr. Sladek,

Thank you for submitting your manuscript to PLOS ONE. After careful consideration, we feel that it has merit but does not fully meet PLOS ONE’s publication criteria as it currently stands. Therefore, we invite you to submit a revised version of the manuscript that addresses the points raised during the review process.

Please provide a point-by-point response to all reviewers' comments. In particular, it would be good to include more examples to illustrate utility, and make sure the software works on the systems it is supposed to work on.

We look forward to receiving your revised manuscript.

Kind regards,

Bostjan Kobe, Ph.D.

Academic Editor

PLOS ONE

“V.S. acknowledges support form projects from Slovak Research and Development  Agency (APVV, https://www.apvv.sk/?lang=en) APVV-19-0376, and from the  Scientific Grant Agency of the Ministry of Education, science, research and sport of the  Slovak Republic and the Slovak Academy of Sciences (VEGA, https://www.minedu.sk/vedecka-grantova-agentura-msvvas-sr-a-sav-vega/) VEGA-2/0031/19 and VEGA-2/0061/20. Y.S. acknowledges support from MEXT Quantum Leap Flagship Program (MEXT Q-LEAP, https://www.jst.go.jp/stpp/q-leap/en/index.html),  Grant Number JPMXS0120330644. 356 This work was supported in part by the Japan Agency for Medical Research and Development (AMED, https://www.amed.go.jp/en/) under Grant Number JP20ae0101047h0001. Part of the calculations were done at the Computing Center of SAS using the infrastructure from Project Nos. ITMS 26230120002 and 26210120002, supported by the Research and Development Operational Program funded by the 361 ERDF. The funders had no role in study design, data collection and analysis, decision to publish, or preparation of the manuscript.”

“V.S. acknowledges support form projects from Slovak Research and Development Agency (APVV, https://www.apvv.sk/?lang=en) APVV-19-0376, and from the Scientific Grant Agency of the Ministry of Education, science, research and sport of the Slovak Republic and the Slovak Academy of Sciences (VEGA, https://www.minedu.sk/vedecka-grantova-agentura-msvvas-sr-a-sav-vega/) VEGA-2/0031/19 and VEGA-2/0061/20. Y.S. acknowledges support from MEXT Quantum Leap Flagship Program (MEXT Q-LEAP, https://www.jst.go.jp/stpp/q-leap/en/index.html) , Grant Number JPMXS0120330644. This work was supported in part by the Japan Agency for Medical Research and Development (AMED, https://www.amed.go.jp/en/) under Grant Number JP20ae0101047h0001. Part of the calculations were done at the Computing Center of SAS using the infrastructure from Project Nos. ITMS 26230120002 and 26210120002, supported by the Research and Development Operational Program funded by the ERDF. The funders had no role in study design, data collection and analysis, decision to publish, or preparation of the manuscript.”

Additional Editor Comments (if provided):

Reviewers' comments:

Reviewer's Responses to Questions

**Comments to the Author**

1. Is the manuscript technically sound, and do the data support the conclusions?

Reviewer #1: Yes

Reviewer #2: Yes

2. Has the statistical analysis been performed appropriately and rigorously? 

Reviewer #1: N/A

Reviewer #2: N/A

3. Have the authors made all data underlying the findings in their manuscript fully available?

Reviewer #1: Yes

Reviewer #2: Yes

4. Is the manuscript presented in an intelligible fashion and written in standard English?

Reviewer #1: Yes

Reviewer #2: Yes

5. Review Comments to the Author

Reviewer #1: This is a well written manuscript about a tool that will be a valuable addition to the scientific community. However, the choice of the single domain protein to identify the protein protein interactions is not a great one. There are other well established PPI proteins that can be used to emphasize the tool's efficiency. This will make the reader clear as to the tool's motive and its relative performance with other existing tools for PPI.

Reviewer #2: Remarks:

1. please, check intall instruction in distributive of plug-in: several files, that is necessary to download from https://www.lfd.uci.edu/~gohlke/pythonlibs/ are absent for Windows. I could not install plug-in in Windows system, Linux - OK.

2. I did not like organisation of manuscript. Many figures that are necessary for understanding the text are located in Supported material. This material are additional material, and the manuscript must be clear without them.

3. to illustrate the work of plug-in "de novo synthesized TIM barrel protein [84]" was used. What PDB code of it?

4. analyzing this protein, several amino acid residues have been identified as important residues for protein-protein interaction. Are there any experimental data that support this conclusions?

Minor remarks:

1. please clarify caption for fig. 3

2. page 5: check link to Figure.

3. Add link to program where prmtop file can be obtained.

6. PLOS authors have the option to publish the peer review history of their article (what does this mean?). If published, this will include your full peer review and any attached files.

Reviewer #1: No

Reviewer #2: No

---

## [Author Response · Author response to Decision Letter 0]

8 Jul 2021

The response is uploaded as a file.

---

## [Editor Report · Decision Letter 1]

12 Jul 2021

pyProGA - A PyMOL plugin for protein residue network analysis

PONE-D-21-15353R1

Dear Dr. Sladek,

We’re pleased to inform you that your manuscript has been judged scientifically suitable for publication and will be formally accepted for publication once it meets all outstanding technical requirements.

Kind regards,

Bostjan Kobe, Ph.D.

Academic Editor

PLOS ONE
---

## [Editor Report · Acceptance letter]

14 Jul 2021

PONE-D-21-15353R1 

pyProGA - A PyMOL plugin for protein residue network analysis 

Dear Dr. Sladek:

I'm pleased to inform you that your manuscript has been deemed suitable for publication in PLOS ONE. Congratulations! Your manuscript is now with our production department. 

Kind regards, 

on behalf of

Professor Bostjan Kobe 

Academic Editor

PLOS ONE